# Effects of Tai Chi on Postural Control in People with Peripheral Neuropathy: A Systematic Review with Meta-Analysis

**DOI:** 10.3390/healthcare11111559

**Published:** 2023-05-26

**Authors:** Wenhui Mao, Ting Wang, Mengzi Sun, Fangtong Zhang, Li Li

**Affiliations:** 1School of Sports Science, Nanjing Normal University, Nanjing 210023, China; 12127@njnu.edu.cn (W.M.); 191502034@njnu.edu.cn (T.W.); 12199@njnu.edu.cn (M.S.); 2School of Sport Science, Beijing Sport University, Beijing 100084, China; zhangfangtong@bsu.edu.cn; 3Department of Health Sciences and Kinesiology, Georgia Southern University, P.O. Box 8076, Statesboro, GA 30460, USA

**Keywords:** Tai Ji, peripheral nervous system diseases, postural control, postural balance

## Abstract

Background: Effects of Tai Chi on people with peripheral neuropathy (PN) are not yet apparent. This systematic review was conducted to evaluate the effects of Tai Chi on postural control in people with PN. Methods: Literature was screened in seven databases for relevant randomized controlled trials. The reports and methodological quality were evaluated. A meta-analysis was performed using RevMan5.4 software. Results: Ten reports were included, involving a total of 344 subjects. The meta-analysis found that Tai Chi therapy for people with PN resulted in a smaller sway area, in the double-leg stance with eyes closed test (SMD = −2.43, I^2^ = 0%), than that observed in the control group, greater distance covered in the six-minute walking test (SMD = −0.46, I^2^ = 49%) and faster performance in the timed-up-and-go test (SMD = 0.68, I^2^ = 50%), than the baseline. Conclusions: Tai chi effectively enhanced dynamic postural control in people with PN. However, no better effects on postural control from Tai Chi than from other rehabilitation approaches were observed in this study. Further high-quality trials are needed to better understand Tai Chi’s effects on individuals with PN.

## 1. Introduction

Peripheral neuropathy (PN) is one of the most common neurological diseases in the elderly population, affecting 20% of the elderly [1]. PN damages the peripheral nervous system from the lower limbs to the spinal cord [2], leading to progressive loss of peripheral sensation, plantar sensitivity, proprioceptive declination, and neuromuscular dysfunction [3]. The loss of muscular mass, power, and strength [4,5] is particularly severe in people with PN in the lower extremity muscles [6]. These deficits predispose people with PN to have a limited range of motion in the lower limb joints, a slow, stiff, unstable gait, easier fatigue, and impaired postural control [7].

Impaired postural control in PN leads to higher risks of falls, with the incidence of falls in PN patients being as much as 20 times higher than in healthy patients [7,8]. Over half of people with PN fall annually at least once [9], leading to decreased physical activity and further postural control impairments [10,11,12,13,14].

Tai Chi, with its low impact and slow motion, is one of the exercise therapies that effectively prevent falls in the elderly [15,16,17]. One meta-analysis showed that Tai Chi could reduce the rate of falls and injury-related falls in older adults by 50% over one year [18]. Tai Chi works synchronously with the hands and legs, offering vital benefits in weight shifting, postural alignments, coordinated movement, and harmonized breathing [19]. It has been postulated that Tai Chi may improve the postural stability of lower limbs [20] and reduce the impact of upper body disturbances on posture [21], which are related to peripheral sensation, proprioception, reaction time in response to postural shift, and neuromuscular function in the lower extremities [22].

Tai Chi has been proposed as an effective therapy for partially restoring neuromuscular function in people with PN. Li and coworkers [14] observed that choosing an appropriate exercise can improve the damaged sensory system, restore activity and postural control, and reduce PN patients’ dependence on others. Tai Chi training restores postural control by reducing the neuropathy total symptom score (TSS) [23], delaying proprioceptive declination by improving somatic sensory sensitivity [24], improving proprioception [25], and reducing the neuromuscular reaction time of the lower extremity muscles [26]. It has been reported that the postural muscles of people who regularly practice Tai Chi can maintain faster reactions during regressive walking [27] and down-step walking [28].

Tai Chi training can improve postural control in people with PN. It is part of the comprehensive rehabilitation of people with PN, closely related to functional rehabilitation. In recent years, more researchers have paid attention to the role of Tai Chi in improving the static postural control [23,29,30,31] and the dynamic postural control [13,24,29,31,32] of people with PN. However, the results are controversial. For example, time in the one-leg standing test (OLST) is a good indicator for static postural control. In some research it was shown that OLST time might be longer after Tai Chi training [23], but other research garnered different results [29,30,31,32].

It is unclear if Tai Chi is effective in the postural control of people with PN. Although there are many studies investigating the effects of Tai Chi on postural control in people with PN, there is a lack of relevant systematic reviews. Previous systematic reviews concentrated on the postural control benefits of Tai Chi, focusing on the non-PN population. Therefore, a new systematic review to determine the efficacy of Tai Chi, comprehensively and systematically, in the postural control of people with PN is necessary to assist healthcare providers, policymakers, and other decision-makers in making their decisions. This review’s specific objective was to evaluate Tai Chi’s effect on postural control in people with PN.

## 2. Materials and Methods

### 2.1. Study Design

This systematic review was registered with INPLASY (International Platform of Registered Systematic Review and Meta-analysis Protocols, INPLASY202340098) and follows applicable Preferred Reporting Items for Systematic Reviews and Meta-Analyses 2020 (PRISMA2020) [33,34]. The review question, constructed according to the population, intervention, comparison, and outcome model was: “Is Tai Chi effective in improving static postural control and dynamic postural control in individuals with PN”. This is a systematic review of published literature and is exempt from IRB approval.

### 2.2. Literature Search Strategy

Details of the search strategy used for each database are shown in the Appendix A. The systematic search was conducted twice on the following databases from inception to 28 April 2023: China National Knowledge Infrastructure (CNKI), China VIP Full-text Database (VIP), Google Scholar, Web of Science databases, Science Direct (SD), Wanfang Data Knowledge Service Platform, and Springer Link. The following keywords and Boolean operators (AND or OR) were used: (Tai Chi OR Tai Chi training OR taijiquan OR t’ai chi ch’uan) ANG (PN OR peripheral neuropathy OR PN OR Polyneuropathy OR disorders of peripheral nerves) ANG (postural control OR equilibrium OR pose control OR balance). Reports in the English language or Chinese language were included.

The reports retrieval process consisted of the following steps: (1) Relevant studies were retrieved from various databases, such as Web of Science, CNKI, and others, using keywords based on the obtained literature; (2) Databases were searched using all relevant keywords, and the titles and abstracts read. If the inclusion criteria were met, the entire paper was searched and read; (3) The references of all included reports were checked for further relevant publications.

### 2.3. Literature Inclusion and Exclusion Criteria

#### 2.3.1. Literature Inclusion Criteria

Participant, intervention, comparison, and outcome (PICO) criteria were set for literature retrieval, inclusion, screening, and exclusion.

##### Participants

Participants with a clinical diagnosis of PN based on symptoms had not practiced Tai Chi in the past four months before the trials. They had an increased vibrating perception threshold (VPT), a decreased nerve conduction velocity (NCV), and engaged in moderate intensity exercise at least once a week. The body mass indices (BMIs) of the participants were between 10 and 35 kg/m^2^.

##### Intervention

The experimental group received Tai Chi alone or with other resistance training. In contrast, the control group received other forms of exercise, such as postural control or routine rehabilitation training. We did not discriminate against different types of Tai Chi intervention since we could not find evidence differentiating the effects of different types of Tai Chi intervention.

##### Comparison

Comparisons between groups and between before and after intervention were conducted. 

##### Outcome Measures

To assess the effects of Tai Chi on static postural control, the One-leg standing test (OLST) and double-leg stance balance with both eyes closed were utilized [30].

To assess improvement in dynamic postural control the Berg balance scale, the ABC balance confidence scale, the six-minute walking distance test (6MWD) (m), and the Timed Up-and-Go Test (TUG) were utilized [11,35].

#### 2.3.2. Literature Exclusion Criteria

Studies were excluded if the full-text article could not be retrieved in English or Chinese, if the articles involved animal experiments, or if the articles were meeting abstracts, case reports, methodological experimental designs, or reviews.

### 2.4. Literature Selection and Data Extraction

Two trained reviewers used a multi-step screening approach to assess the studies independently and recorded the main reasons for exclusion. The author, publication year, and title were reviewed in the first screening to determine which studies to include. In the second screening, titles and abstracts were reviewed to exclude studies that did not meet the inclusion criteria. In the final screening, the full texts were reviewed to determine the outcome indicators and other relevant data, such as author, publication year, sample size, selection criteria, intervention measures, frequency and intensity, intervention duration, follow-up time, causes of loss to follow-up, results, and conclusions.

Disagreements were resolved through discussion. The final decision was made by a third author if consensus could not be reached through discussion. This data was then extracted for use in the meta-analysis by one author independently and was confirmed by another author.

In addition, the following literature characteristics of each included report were documented: reasons for duplicative systematic reviews, number of actual applications versus previous reviews, experimental methods of inclusion, demographic data and characteristics. Heterogeneity and subgroup analysis of the main studies were also documented, along with review conclusions as to whether Tai Chi was effective in improving the balance function of people with PN.

### 2.5. Assessment of Methodological Quality of Studies

Methodological quality evaluation for papers included in the meta-analysis was conducted by two researchers independently, using the Cochrane Collaboration’s tool to assess the risk of bias (updated 2022) [36]. The evaluation adopted the following criteria: (1) Random sequence generation (selection bias); (2) Allocation concealment (selection bias); (3) Blinding of participants and personnel (performance bias); (4) Blinding of outcome assessment (detection bias); (5) Incomplete outcome data (attrition bias); (6) Selective reporting (reporting bias); (7) Other bias.

Papers were classified into three levels based on the potential for bias. A paper was rated as grade A if it met all of the quality standards in the Cochrane Collaboration’s tool for assessing the risk of bias and the potential for bias was low. A paper was rated as grade B if it met some of the standards and the potential for bias was moderate. A paper was rated as grade C if it did not meet any of the standards, and the potential for bias was high. After independently evaluating the quality of each paper, the researchers discussed their assessments and decided whether to include or exclude the paper from the analysis. A third author made the final decision if this was required.

### 2.6. Statistical Analysis

The meta-analysis was performed using RevMan 5.3.0 (Cochrane Collaboration, London, UK) for data processing, merging of data, performing a sensitivity analysis on the outcome indicators, determining risk of bias, and the drawing of forest plots. Continuous variables were analyzed in a meta-analysis using the standard mean difference (SMD) with a 95% confidence interval (CI), calculated as the effect size of the outcome. SMD was interpreted as follows: small effect, 0.2–0.6; medium effect; 0.6–1.2, large effect, 1.2–2.0; and very large effect, >2.0 [37]. Heterogeneity was tested by means of the Higgins I^2^, with I^2^ values of ≤25% representing no to low heterogeneity, 25% < I^2^ ≤ 50% representing moderate heterogeneity, 50% < I^2^ ≤ 75% representing high heterogeneity, and I^2^ > 75% representing very high heterogeneity. If the I^2^ value was less than 50%, a fixed-effects model was used for combination, and if the I^2^ was greater than, or equal to, 50%, a random-effects model was used [24,38]. Sensitivity analysis was performed by eliminating papers stepwise to determine if a single study significantly affected the heterogeneity. The level of statistical significance was set at α < 0.05.

## 3. Results

### 3.1. Literature Search Results

Following title/abstract screening, the rest of the studies were carefully screened for inclusion, and there were still 359 articles. Only 9 studies [23,24,29,30,31,32,39,40,41] met our inclusion criteria and were included in the meta-analysis abstract. A report was included by citation searching [13] (Figure 1). All the articles included were published between 2006 and 2019. None of the included studies had conflicts of interest. In the meta-analyses, 199 male and 145 female subjects were included., The average age of the subjects was 67.6 ± 6.5 years. The regularity of the interventions ranged from one, two, or three times per week to seven times per week, and the times of each intervention ranged from 45 min [39] to 90 min [13], although most studies were set at 60 min [23,24,29,30,31,32,40,41]. The duration was from 8 weeks to 24 weeks. As a control intervention, the Tai Chi exercises mainly included 24-style Tai Chi [40,41], Yang’s Tai Chi [24,30,32], and self-designed 27-style Tai Chi [13], which emphasizes slow intentional movements, weight transfer between the lower body and controlled breathing. All studies reported specific methods of Tai Chi training, taught and guided by experienced Tai Chi experts [13,23,24,29,30,31,39,40,41], and the training site environment, including nursing homes [30], was recorded by the lecturer to evaluate the compliance of the Tai Chi intervention [24]. A total of 6 experimental studies had been carried out in the United States [13,24,30,31,32,39].

### 3.2. Basic Characteristics of Included Studies

A total of 8 articles [13,23,24,29,30,31,32,39] were in English and two articles [40,41] were in Chinese. The publication years were 2006 to 2019, and the sample size ranged from 8 to 68, with 344 people were included (Table 1).

### 3.3. Literature Quality Evaluation

The quality of 1 study, Patricia AQ, was evaluated as “A”, and the quality of the remaining 9 studies as “B”. Five papers were low risk, 5 papers could not be judged as low risk or high risk in random sequence generation, Five papers introduced the specific methods of allocation concealment and 7 papers had no the allocation concealment scheme. Nine studies had no relevant information about the blinding of participants and personnel and blinding of outcome assessment. Eight articles reported complete data and missing data, with reasons described. Seven articles were low risk in selective reporting and 7 papers could not be judged as low risk or high risk (Figure 2 and Figure 3).

### 3.4. Meta-Analysis of Measured Outcomes

#### 3.4.1. Static Postural Control

Four meta-analysis studies [23,29,30,31] reported on OLST in the Tai Chi and control groups. As shown in Figure 4A, the random-effects model was used, because the heterogeneity between studies was moderate (*p* for heterogeneity = 0.10, I^2^ = 52%). There were no significant differences in the times for the OLST between the Tai Chi and the control groups (SMD = 0.25, 95% CI: −0.29~0.79, *p* = 0.37). Due to the moderate heterogeneity between studies, we performed a sensitivity analysis. Firstly, a lower-quality study was excluded [27]. This exclusion did not change the outcome and statistical heterogeneity (SMD = 0.07, I^2^ = 54%, *p* = 0.84). Secondly, a study with a small sample size was excluded [40], but this exclusion did not change the results and statistical heterogeneity (SMD = 0.37, I^2^ = 55%, *p* = 0.20). Finally, excluding one study with results significantly different from several others [30] did not change the results and statistical heterogeneity (SMD = 0.45, I^2^ = 53%, *p* = 0.13). A total of three sensitivity analyses were performed and similar results were obtained, indicating that the results presented in this paper are stable.

Figure 4B shows that there was no heterogeneity between studies (*p* for heterogeneity = 0.97, I^2^ = 0%) and the fixed-effect model was used. Compared with the control group, the swaying area of the Tai Chi group after training was significantly reduced (SMD = −2.43, 95% CI: −3.07~−1.80, *p* < 0.00001).

#### 3.4.2. Dynamic Postural Control

Three studies [13,31,39] reported balancing scale scores for both groups before and after the Tai Chi training. As shown in Figure 5A, there was no heterogeneity between studies and the fixed-effect model was used (*p* for heterogeneity = 0.89, I^2^ = 0%). The balance scale scores were not different between the Tai Chi group and the control (SMD = −0.08, 95% CI: −0.38–0.53, *p* = 0.74).

There was no heterogeneity in the two studies [24,29] for 6MWD and the fixed-effect model was used (*p* for heterogeneity = 0.95, I^2^ = 0%). The distances of the 6MWD did not differ between the Tai Chi and control groups (SMD = 0.10, 95% CI: −0.46~0.65, *p* = 0.73) (Figure 5B).

Three RCTs [13,24,31] reported on comparisons of TUG between the Tai Chi group and the control group. The results showed no heterogeneity among studies and a fixed-effects model was used (*p* for heterogeneity = 0.69; I^2^ = 0%). There were no differences in TUG time between the Tai Chi group and the control group (SMD = −0.19, 95% CI: −0.66~0.29, *p* = 0.44) (Figure 5C).

Three studies [24,29,32] reported on the 6MWD in both groups before and after training. The distance of 6MWD before and after training in the Tai Chi group was significantly longer (SMD = −0.46, 95% CI: −0.94~0.01, *p* = 0.06). The heterogeneity between studies was low and a fixed-effect model was used (*p* for heterogeneity = 0.14, I^2^ = 49%) (Figure 6A).

Four studies [13,24,31,32] reported on TUG performed in the Tai Chi group before and after training. The results showed middle level heterogeneity between studies and a fixed-effects model was used (*p* for heterogeneity = 0.11, I^2^ = 50%). The TUG time of the Tai Chi group was significantly shorter after training (SMD = 0.68, 95% CI: 0.28~1.09, *p* = 0.0010) (Figure 6B).

## 4. Discussion

This systematic review with meta-analysis aimed to assess the effect of Tai Chi intervention on postural control among people with PN. The main findings from this review suggest that Tai Chi had a high clinical effect (SMD = −2.43) on the improvement of static postural control, as measured by changes in the sway area test of double-leg stance with eyes closed, compared with the control group, and a small clinical effect (SMD = −0.46, 0.68) in dynamic postural control, as measured by changes in 6MWD and TUG, compared with baseline. However, Tai Chi had no better effect than other rehabilitation approaches of the control group on OLSD and dynamic postural control.

### 4.1. Basic Characteristics of Included Studies

The sample size included in this meta-analysis included 199 males and 145 females, mainly aged over 60 years. The duration of intervention was mainly 60 min and ranged from 8 weeks to 24 weeks. The types of Tai Chi were 16, 24, and 27-style. Types of intervention in the control group included aerobics, stretching, routine rehabilitation, balance training, medication, and 24-style Tai Chi with resistance training.

### 4.2. Included Literature Quality

This study reviewed 10 [13,23,24,29,30,31,32,39,40,41] RCTs containing evidence on the effects of Tai Chi on postural control in people with PN. However, these RCTs had methodological, reporting, and quality of evidence limitations. The results of the methodological quality assessment showed that all included RCTs had more than one item that needed to be met. Therefore, these RCTs [24] were considered to be of low methodological quality. The suboptimal quality of the method was due to the fact it did not conform to the method of random sequence and allocation concealment. As for the report quality, the evaluation results showed that 8 RCTs [23,24,29,30,31,32,40,41] reported all items, while other studies [13,39] were missing report content to varying degrees. The undesirable quality of reporting was attributed to search, risk of bias across studies, and insufficient reporting of additional analyses. In the quality of evidence, 10 outcomes were included; only 1 [41] was of high quality, 8 [13,23,29,30,32,39,40,41] were of moderate quality, and 1 [24] was of low quality. The risk of bias was the most common factor leading to the downgrading of evidence, followed by inconsistency, imprecision, publication bias, and discontinuity.

The impact of research conclusions on the quality of the literature is as follows. Firstly, due to the low quality of the included randomized controlled trials, most researchers came to relatively definite conclusions. Higher-quality RCTs help provide scientific evidence. This review observed that the overall methodological quality, reporting quality, and evidence quality of the included RCTs varied, indicating limitations in the reliability of the conclusions of these RCTs. Secondly, scholars mostly used the term “Tai Chi” to represent all types of Tai Chi. There are various types of Tai Chi, including “Chen, Yang, Wu, Sun,” and others. Different types of Tai Chi have different effects on rehabilitation. This study did not further analyze and compare different types of Tai Chi. A subgroup analysis for different types of Tai Chi is necessary. Furthermore, the diversity of Tai Chi, including differences in training style, exercise form, frequency, and duration, may be a source of heterogeneity in RCTs. This also requires further analysis.

### 4.3. Static Postural Control

Both Double-leg stances with eyes closed and OLST were used to evaluate the improvement of static postural stability of Tai Chi in people with PN [24,30,40,41]. The Tai Chi group exhibited better, and more clinical, reduction of sway area in a double-leg stance with eyes closed than the control group, in line with previous studies [39,41]. However, Tai Chi did not exhibit a better effect on OLST than other treatments. Tai Chi improved the time of OLST in non-diseased older adults [42], but limited improvement in people with chronic diseases [43].

There may be several reasons. First, compared to the double-leg stance with eyes closed test, OLST is a more challenging stability test, due to the small support surface in postural control research [13]. The resolution of the OLST test is not enough to assess the effect of low-intensity intervention. Second, the performance of OLST is closely related to lower limb muscle strength. Tai Chi is a type of moderate- to low-intensity aerobic weight-bearing exercise [44]. Compared with resistance exercise, Tai Chi had no advantage in improving lower limb muscle strength. Tai Chi frequency in Richerson [30] and Quigley [31] was once a week, in both Tsang [29] and Ahn [23] the frequency was twice per week. These frequencies of Tai Chi to assess effects on static postural control in people with PN are low. On the other hand, the heterogeneity was high and the clinical effect low (SMD = 0.25) in the OLST meta-analysis. This indicates that multiple factors may have led to this negative result in regard to the Tai Chi effect on OLST, such as different total sports load of Tai Chi and the control, intensity, age, and course of disease.

The result of the double-leg stance with eyes closed in this study indicated that Tai Chi may enhance postural control and reduce the risk of falls. A recent study showed that Tai Chi could improve lower extremity strength [35]. For people with PN, significant improvements in muscle strength and functional performance after 18 weeks of Tai Chi training in [32]. Still, this may be because greater intervention intensity is needed to improve limb strength and static postural control in people with PN. So, the effects of Tai Chi on static postural control in people with PN needs to be further investigated [31].

### 4.4. Dynamic Postural Control

A previous study reported that people with PN exhibit abnormal walking patterns, including increased loading times on the heel and metatarsal and decreased loading on the hallux [35]. The Balance Scale, TUG, and 6MWD are valid assessment tools for functional mobility and postural control and have been shown to accurately predict repeated falls in people with PN [11,45]. This study observed no significant difference in the indicators between the two groups after intervention training. The clinical effects were low (SMD = −0.08, 0.10, −0.19) in the assessments. The three indicators reached a consensus for the same effect of Tai Chi on dynamic postural control in people with PN compared with the control group. This suggests that Tai Chi does not have a better effect on dynamic postural control than other rehabilitation approaches. As discussed earlier, a comparison of different rehabilitation treatments is affected by multiple factors. Accuracy RCTs are needed to clarify the effects of Tai Chi under different conditions.

The other assessment of the Tai Chi effect on dynamic control in this study showed that Tai Chi training improved indicators of dynamic postural control such as walking faster and farther after training than baseline [13,24,31,32]. The observations of this study provide stable and consistent evidence that Tai Chi can improve dynamic postural control in people with PN. The clinical effects were small and medium, respectively (SMD = −0.46, 0.68). This suggests that Tai Chi is still an alternative rehabilitation treatment.

The effect of Tai Chi on postural control has been confirmed in older adults [46] and people with postural control dysfunction caused by various diseases, such as Diabetes mellitus [47], Parkinson’s disease [48], and Stroke [49]. Tai Chi can improve the Balance Confidence Scale scores of the elderly [50,51], increase 6MWD [43,52], and shorten the time of TUG [43,53] in people with chronic diseases. In addition, in the aging population, modified Chen-style Tai Chi was found to be more effective than 24-style Tai Chi in enhancing health-related parameters [54].

The mechanisms of the results are not very clear. Some studies showed that Tai Chi effectively improves postural control by focusing on transferring the center of mass (COM) and stabilizing the ankle swing, enhancing the function of the sensorimotor system and knee joint extension strength [55,56]. Other studies showed that Tai Chi can improve fall-related factors, such as fear of falling [57], and musculoskeletal strength [58]. Tai Chi also improved plantar sensation [30] and peripheral nerve conduction properties [11] in older adults. Pei and his colleagues [59] observed that Tai Chi practitioners showed better hand–eye coordination and motor control.

Tai Chi is well-tolerated, safe, economical, and enjoyable for older adults [34]. The results of this study provide further support for the benefits of Tai Chi in improving postural control in people with PN. However, this study did not obtain enough evidence for better postural control from practicing Tai Chi than from other rehabilitation approaches.

### 4.5. Limitation

This study has the following limitations that must be considered: (1) The scales used for outcome indicators in the 10 studies were subjective, leading to differences in the operation of the same scale evaluation, thereby reducing the reliability of the meta-analysis; (2) The treatment durations in the included articles ranged from 8 to 24 weeks, and factors, such as training time, training period, method of determining PN, and outcome indicators, may have impacted the results. With a small number of RCTs included, the study was unable to conduct further stratified analysis based on indicators such as age, gender, and intervention methods; (3) The term “Tai Chi” was used in this study to represent all types of Tai Chi without specifying the different types, leading to a lack of information on specific Tai Chi styles. As a result, this review cannot recommend using a particular type of Tai Chi.

## 5. Conclusions

Tai chi effectively enhanced dynamic postural control in people with PN. However, no better effects on postural control from Tai Chi than those derived from the other rehabilitation treatments were observed in this meta-analysis. Despite this, the limited quality of the studies suggests there is a need for further high-quality trials to confirm the long-term benefits of Tai chi on individuals with PN.

## Figures and Tables

**Figure 1 healthcare-11-01559-f001:**
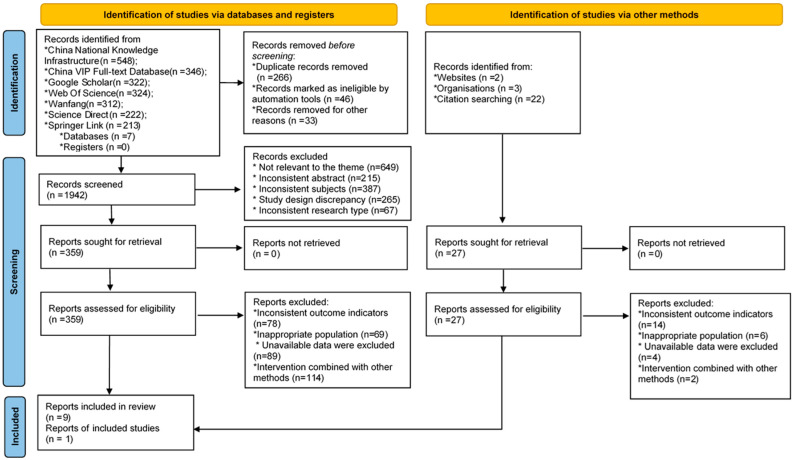
Flow diagram of the screening process.

**Figure 2 healthcare-11-01559-f002:**
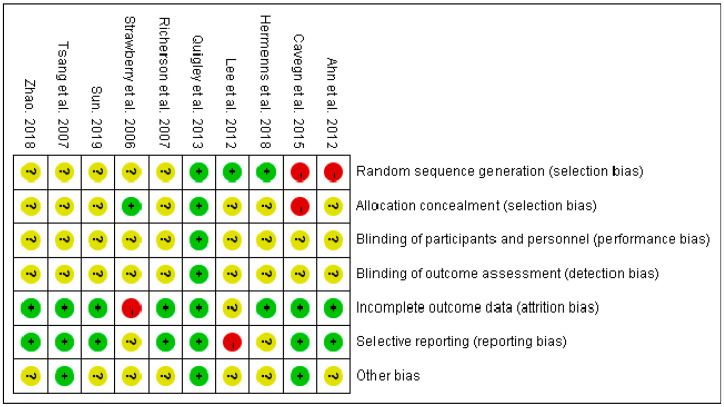
Risk of bias assessment of included studies [13,23,24,29,30,31,32,39,40,41]. (+, meet the criteria, −, not meet the criteria, ?, not clear).

**Figure 3 healthcare-11-01559-f003:**
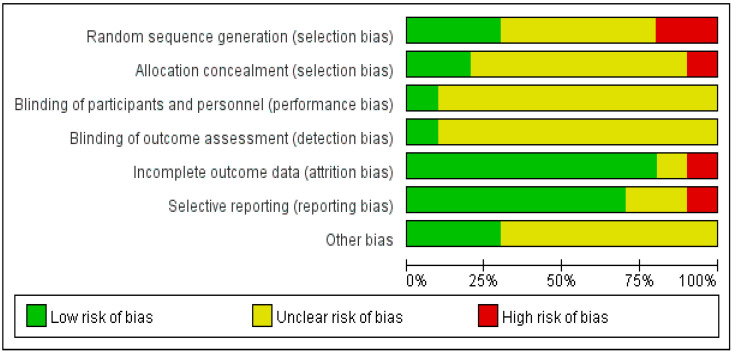
The overall risk of bias assessment of included studies.

**Figure 4 healthcare-11-01559-f004:**
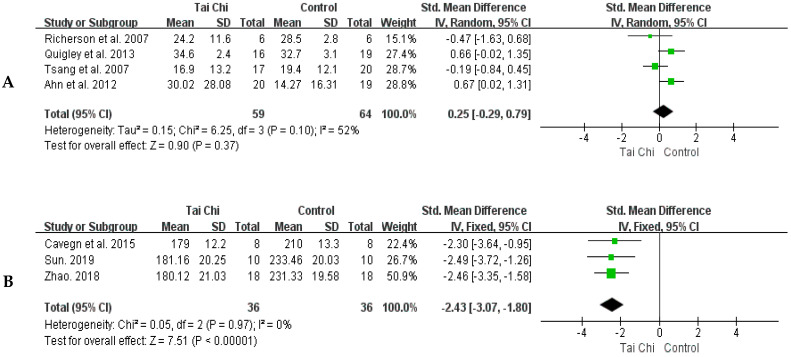
Forest plots of comparison between the Tai Chi group and control group for the time of OLST (**A**) and sway area of double-leg stance with eyes closed (**B**) [23,24,29,30,31,40,41]. The green box and the black rhombus represent results of the individual studies and the combined results, respectively.

**Figure 5 healthcare-11-01559-f005:**
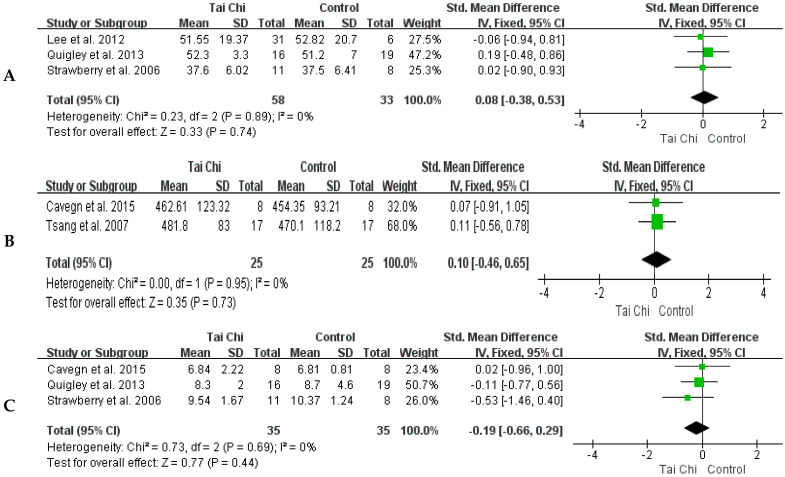
Forest plots of comparison between the Tai Chi group and control group for the score of the Balance Scale (**A**), the distance of 6MWD (**B**), and time of TUG (**C**) [13,24,29,31,39]. The green box and the black rhombus represent results of the individual studies and the combined results, respectively.

**Figure 6 healthcare-11-01559-f006:**
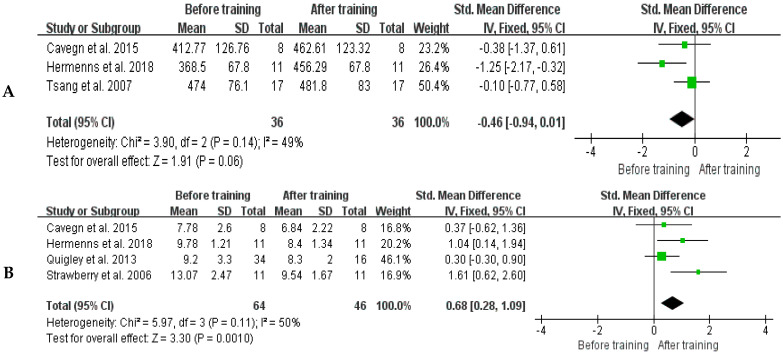
Forest plots of comparison between before and after Tai Chi training for the distance of 6MWD (**A**) and TUG (**B**) [13,24,29,30,31,32]. The green box and the black rhombus represent results of the individual studies and the combined results, respectively.

**Table 1 healthcare-11-01559-t001:** Basic characteristics of included studies.

Author	Location	Participants			Tai Chi Group			The Intervention of the Control Group	Duration	Outcome
		Age (Duration of Disease, Year)	Sample Size	Gender (M/F)	Intervention	Duration per Time (min)	Frequency (Times/Week)			
Tsang et al., 2007 [29]	Australia	T66 ± 8 (8.5 ± 2.0) C65 ± 8 (9.0 ± 0.7)	T18 C20	T14/4 C14/6	Tai Chi	60	2	Aerobics, gentle stretching, routine nursing	16 weeks	①; ⑤
Richerson et al., 2007 [30]	United States	T72.92 ± 5.21 C74.50 ± 7.72	T12 C6	T12/0 C6/0	Yang Tai Chi	60	1	Routine rehabilitation	24 weeks	①
Ahn et al., 2012 [23]	Korea	T66.05 ± 6.42 (7.66 ± 2.51) C62.73 ± 7.53 (7.97 ± 2.64)	T20 C19	T12/8 C8/11	Tai Chi	60	2	Routine rehabilitation	12 weeks	①
Quigley et al., 2013 [31]	United States	T68.4 ± 9.3 (7.9 ± 4.2) C_1_67.6 ± 10.6 (8.0 ± 4.8) C_2_67.5 ± 10.2 (10.1 ± 5.4)	T34 C_1_34 C_2_31	T5/29 C_1_4/30 C_2_6/25	Tai Chi	60	1	C_1_: Routine rehabilitation C_2_: Balance training	10 weeks	②; ⑥
Cavegn et al., 2015 [24]	United States	T65.5 ± 7.4 (18.63 ± 9.21) C63.8 ± 5.7	T8 C8	T2/6 C2/6	Yang Tai Chi	60	3	Routine rehabilitation	8 weeks	⑤; ⑥
Zhao 2018 [40]	China	T57.51 ± 6.85 (6.17 ± 3.65) C56.92 ± 5.61 (5.64 ± 3.81)	T18 C18	T14/4 C15/3	24-style Tai Chi	60	7	Routine rehabilitation	12 weeks	③;
Sun 2019 [41]	China	T58.78 ± 6.21 (7.96 ± 2.45) C_1_57.86 ± 5.66 (7.75 ± 2.30) C_2_58.55 ± 6.15 (7.28 ± 2.65)	T10 C_1_10 C_2_10	T10/0 C_1_10/0 C_2_10/0	24-style Tai Chi	60	7	C_1_: Medication C_2_: 24-style Tai Chi with resistance training	16 weeks	①;
Strawberry et al., 2006 [13]	United States	T77.6 ± 7.0 C77.5 ± 5.5	T11 C8	T10/1 C7/1	27-style Tai Chi	90	5	Routine rehabilitation	12 weeks	②; ⑥
Hermenns et al., 2018 [32]	United States	T74.5 (13.3)	T12	T7/5	Yang Tai Chi	60	2	Routine rehabilitation	16 weeks	①; ⑥
Lee et al., 2012 [39]	United States	T70.2 ± 9.81 C72.7 ± 7.6	T20 C17	T17/3 C14/3	16-style Tai Chi	45	1	Routine rehabilitation	8 weeks	④

① One-leg standing test (OLST) (s); ② double-leg stance balance with eyes closed (s); ③ Berg balance scale; ④ ABC balance confidence scale; ⑤ 6-min walking distance test (6MWD); ⑥ Timed Up-and-Go Test (TUG); C, control group; T, Tai Chi group; M, male; F, female.

## Data Availability

Datasets analyzed for the current study are available from the first author up on reasonable request.

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
