# Peer review of "Effects of Tai Chi on Postural Control in People with Peripheral Neuropathy: A Systematic Review with Meta-Analysis"

_healthcare, 2023, doi:10.3390/healthcare11111559_

Round 1
Reviewer 1 Report
Dear Authors:
Thank you for allowing me to review your manuscript. You have done an excellent job. However, I would like to make some observations in order to improve the manuscript:
-Introduction: Adequate and well framed the subject under study. However, in the results and discussion you mention aspects such as the types of tai chi. Perhaps if in the introduction you would explain about the types of TAI CHI it would improve the general understanding of the manuscript. You correctly point out that the types of TAI CHI is an aspect that limits the results of your work.
-Materials and Methods:
Literature search strategy: This section needs to be improved. When reference is made to search terms, it should be specified whether they were mesh terms or free terms. The final search equation should be indicated. It is advisable to indicate who carried out the searches (how many people and if they are different from those who carried out the literature selection).
It is not indicated how possible disagreements were resolved in this phase (whether by consensus or by a third person who acted as referee, as you do in the section Assessment of methodological quality of studies).
You should also explain what were these important sources that you point out in the flowchart.
-It would be advisable to explain each Outcome measures. Some of these variables are not known to the readers. This may help to improve the understanding of the results of the study.
In table 1 you should correct Unite states by United states and to improve the comprehension of the table, you should add a footnote or some explanatory note exploding the acronyms T and C (I understand that T=group Tai Chi and C= Group Control).
Discussion: This sentence in the discussion does not correspond to the results of your study:
The meta-analysis showed that Tai Chi exercise improves postural control as measured by changes in OLST, sway area test of double-leg stance with eyes closed, Balance Confidence Scale, 6MWD, TUG, and PPDT as outcome measures.
Tai chi has not been shown to be useful on the basis of all outcome variables. You should therefore be more cautious and modify this based on the results of your study.
Good job. Best regards
Author Response
Reviewer: -Introduction: Adequate and well framed the subject under study. However, in the results and discussion you mention aspects such as the types of tai chi. Perhaps if in the introduction you would explain about the types of TAI CHI it would improve the general understanding of the manuscript. You correctly point out that the types of TAI CHI is an aspect that limits the results of your work.
Authors’ response: Thank you. At present, the main difference between the types of Tai Chi lies in the number of moves. No studies have demonstrated the effects of different types of Tai Chi on rehabilitation outcomes. In addition to the statement in the limitation section, we have added a statement in section 2.3.1 at the end of the intervention paragraph to discuss this information.
-Materials and Methods:
Reviewer: Literature search strategy: This section needs to be improved. When reference is made to search terms, it should be specified whether they were mesh terms or free terms. The final search equation should be indicated. It is advisable to indicate who carried out the searches (how many people and if they are different from those who carried out the literature selection).
Authors’ response: Thank you very much for your advice. We searched the database again to ensure that new studies met our included criteria. Then we provided supplementary material on the Literature search strategy this time.
And in lines 87 – 96 of the revised manuscript we updated the search time, added the details of the search equation, and filter. “Details of the search strategy used for each database are shown in the supplementary material. The systematic search was conducted twice on the following databases from inception to 28th April 2023: China National Knowledge Infrastructure (CNKI), China VIP Full-text Database (VIP), Google Scholar, Web of Science databases, Science Direct (SD), Wanfang Data Knowledge Service Platform, Pubmed, and Springer Link. The following keywords, Boolean operators (AND or OR) were used: (Tai Chi OR Tai Chi training OR taijiquan OR t'ai chi ch'uan) ANG (PN OR peripheral neuropathy OR PN OR Polyneuropathy OR disorders of peripheral nerves) ANG (postural control OR equilibrium OR pose control OR balance). Reports in the English language and Chinese language were included.”
In lines 130 – 139, more details were added to the selection and data extraction process. “Two trained reviewers used a multi-step screening approach to assess the included studies independently and recorded the main reason for exclusion. The author, publication year, and title were reviewed in the first screening to determine including studies. In the second screening, titles, and abstracts were reviewed to exclude studies that did not meet the inclusion criteria. In the final screening, the full texts were reviewed to determine the outcome indicators and other relevant data such as author, publication year, sample size, selection criteria, intervention measures, frequency and intensity, intervention duration, follow-up time, causes of loss to follow-up, results, and conclusions. Disagreements were resolved through discussion. And the final decision was made by a third author if consensus can’t be reached by discussion. This data was then extracted for use in the meta-analysis by one author independently and was confirmed by another author.”
Reviewer: Studies numbered 14 and 33 come from references. We used the mesh term “Tai Chi” to search.
It is not indicated how possible disagreements were resolved in this phase (whether by consensus or by a third person who acted as referee, as you do in the section Assessment of methodological quality of studies).
Authors’ response: Thank you. In lines 160 – 162, we added the description of the procedure. All disagreements were solved by consensus. But we kept the original procedure. “The procedure of the assessment was the same as that of Literature selection and data extraction.”
Reviewer: You should also explain what were these important sources that you point out in the flowchart.
Authors’ response: Thank you. The flowchart was updated according to your advice and PRISMA 2020.
Reviewer: -It would be advisable to explain each Outcome measures. Some of these variables are not known to the readers. This may help to improve the understanding of the results of the study.
Authors’ response: Thank you. In lines 119 – 124, we added some explanation of those indicators. “The following measures were used to assess the effects of Tai Chi on postural control: (1) One-leg standing test (OLST), and (2) double-leg stance balance with both eyes closed (s) to assess the improvement of static postural control[35]; (3) Berg balance scale, (4) ABC balance confidence scale, (5) 6-minute walking distance test (6MWD) (m) to assess dynamic postural control, and (6) Timed Up-and-Go Test (TUG) (s) to assess the improvement of dynamic postural control [11, 12, 40].”
Reviewer: In Table 1 you should correct Unite states by the United States and to improve the comprehension of the table, you should add a footnote or some explanatory note exploding the acronyms T and C (I understand that T=group Tai Chi and C= Group Control).
Authors’ response: Sorry for this. Unite states have been corrected and explanatory notes have been added in Table 1.
Discussion:
Reviewer: This sentence in the discussion does not correspond to the results of your study:
The meta-analysis showed that Tai Chi exercise improves postural control as measured by changes in OLST, sway area test of double-leg stance with eyes closed, Balance Confidence Scale, 6MWD, TUG, and PPDT as outcome measures.
Tai chi has not been shown to be useful based on all outcome variables. You should therefore be more cautious and modify this based on the results of your study.
Authors’ response: Thank you for pointing this out. “OLST” was deleted from this sentence. The sentences were reorganized and tried to follow the results of our study (Line 316 – 23).
“The main findings from this review suggest that Tai Chi has a high clinical effect (SMD=-2.29) on the improvement of static postural control as measured by changes sway area test of double-leg stance with eyes closed compared with the control group, and a medium clinical effect (SMD=-0.59, 0.64) in dynamic postural control as measured by changes in Balance Confidence Scale, 6MWD, TUG compared with baseline.”
Reviewer 2 Report
GENERAL COMMENTS:
Firstly, I would like to thank the opportunity to review this systematic review with meta-analysis that aimed to evaluate the effects of TAI CHI on postural control of people with diabetic peripheral neuropathy.
The general procedures and those aspects related with statistical analysis are appropriated and well-conducted. Introduction included a good justification of the study. This is a well-written manuscript, with practical applications for clinicians and practitioners focused on Tai Chi for people with diabetic peripheral neuropathy.
The date of the database’s inception (February 2022) is an important aspect to take into account. If this systematic review is published, it would be appropriate that it is as up to date as possible (at least, up to the end of the last year).
However, the information of the rest of sections needs to be re-organized to maintain the same structure in the entire manuscript and be consistent.
Some parts of the manuscript need to be explained with more details (mainly in Methods).
Results and Discussion sections are confusing in some parts, and lack of fluency. These sections are difficult to understand.
All tables and figures should be deeply revised (for punctuation, format and clarity).
In terms of format and writing, punctuation should be checked in the entire manuscript. References in the text do not maintain the same format (sometimes, there is a white space before and after the brackets, and this is not consistent throughout the manuscript).
SPECIFIC COMMENTS:
INTRODUCTION
It is a well-structured Introduction section. However, it contains too many parts with specific information of studies, which increases the extension of this section. It is recommended to include information in a general way, and to avoid giving details from studies. Lines 58 to 74 and 75 to 84 are confusing. These paragraphs have information about different aspects that can be improved with Tai Chi, without being linked. This fact makes that these paragraph lacks of fluency. It is recommended to re-organize this information according to the key aspects (those really related with postural control such as Time up and go, etc).
Line 85: This sentence is unclear, please reword it.
Line 87: “the efficacy of Tai Chi on people with DPN”, but, what exactly? postural control of these people?
Also, it would be appropriate to focus on the importance of postural control for this pathology in this section, and those test/functions that authors considered as postural control.
METHODS
PubMed is not a database.
Why February 2022? why not the entire 2022 year? It is recommended to consider this aspect to have an updated systematic review. If this systematic review is published, it would be useful to be as up to date as possible (at least, up to the end of the last year).
Lines 112-113: Since this is a systematic review, it would be more appropriate include “studies that recruited participants according to the following criteria…” because this work did not really include participants.
The same for the exclusion criteria.
Intervention: why 8 weeks? to include a reference would be useful to understand the minimal duration that authors proposed.
Outcome measures: I do not understand why authors considered the PPDT test as a postural control test. This test is highly different from the rest of test considered (all of which inform about gait function or postural stability). Re-consider this inclusion criteria to maintain the coherence of the test considered as eligible.
Also, all tests included should be explained with more details in order to be able to understand the Results section.
Study design: RCTs need to be explained the first time it appears in the text (the abbreviation has been explained in a posterior paragraph, please, check it).
Lines 138 – 144: This paragraph does not add any information. Exclusion criteria cannot be the opposite to the inclusion criteria. If there is no exclusion criteria to highlight, it is recommended to delete this paragraph.
Data extraction procedure should be detailed. What researches? was this screening carried out independently? what if disagreement among both researches? who verified this information? Also, it should be detailed what information was extracted: causes of loss to follow-up, results, conclusions, etc is not correct. What data was extracted for the MA?
Lines 154 – 169: On what reference are based the procedure to established the risk of bias classification? In this sense, many tools specifically designed to evaluate risk of bias of RCT are available in the literature (i.e. those tools proposed by Cochrane). Please, consider to use it. Pedro scale has 11 items (10 items for the score), why authors used only 8 items? Again, what happened in case of disagreement? it is not clear in the text. Please, re-arrange information included to explain the risk-of-bias evaluation. A reference is needed.
RESULTS
This section needs to be re-structured and well-organized. Now, text, tables and figures are difficult to understand and follow because the lack of organization.
Figure 1: Quality of the figure is poor. Some sentences in the boxes begin with lowercase, please, check it. In many brackets, there are white spaces that should be checked. Also, it is recommended to modify figure letter type to maintain the same format in the entire manuscript. There are abbreviation without explaining.
It is recommended not to start the RESULTS heading with a figure that has not already been referenced in the text. Instead, add the referencing text.
This text should not duplicate the figure 1. Please, modify it and include the essential data or trends to highlight.
Lines 197 – 198: This information has been included in the previous paragraph, delete it.
Basic characteristics of included studies: How many females and males? age? duration of interventions? type of exercises as control intervention? Despite the fact that this information is detailed in the table, authors should highlight those important aspects in the text. What about the locations of the investigations? It is useful to know where these experimental studies have been carried out (highlighting the most repeated locations, for example).
Table 1 needs a deep revision of the punctuation and other important aspects (spaces between signs, abbreviations without explaining, etc). Take into account that Table should be understandable by itself.
Line 203: the first sentence can be deleted (since it belongs to Methods). Also, it is recommended to simplify this paragraph, now it is difficult to read (and it is duplicating information of Figures 2 and 3).
Lines 233-236: This information should be moved to Methods section.
Lines 236 – 238: Again, this information belongs to statistical analysis.
Lines 233 – 247: The entire paragraph should me rearrange to make easier to follow. Authors should include only that information about results, by avoiding to include information about what statistical analysis were completed or what are the meaning of the test or results.
It is recommended to join figures 4 to 11 in an only figure with different parts (because all these figures represent Forest plots of the MA). It would be easier to understand and paper structure would improve.
DISCUSSION
This should be organized by following the same information previously included in the manuscript. If Methods section did not include postural control test as “dynamic” or “static”, why authors included these subheading in the discussion?
Please, try to maintain the consistency in the entire manuscript. Therefore, the entire discussion should be re-organized to be consistent with the previous part of the manuscript. Now, it is difficult to understand.
Format should be revised. Punctuation should be checked in the entire manuscript. References in the text do not maintain the same format (sometimes, there is a white space before and after the brackets, but it is not consistent throughout the manuscript). Please, check it.
Author Response
Reviewer: The date of the database’s inception (February 2022) is an important aspect to take into account. If this systematic review is published, it would be appropriate that it is as up to date as possible (at Authors’ response: least, up to the end of the last year).
Authors’ response: Thank you very much. The first search time was far from now indeed. We searched the database again to make sure that new studies met our included criteria. No new study met the included criteria. Then we provided supplementary material on the Literature search strategy this time. And lines 87 – 95 of the revised manuscript. We updated the search time and also the flowchart.
Reviewer: However, the information in the rest of the sections needs to be reorganized to maintain the same structure in the entire manuscript and consistency.
Authors’ response: Thank you. Most information in the rest of sections has been reorganized.
Reviewer: Some parts of the manuscript need to be explained in more detail (mainly in Methods).
Authors’ response: Thank you. We followed the specified advice and this advice to add more details.
Reviewer: Results and Discussion sections are confusing in some parts, and lack of fluency. These sections are difficult to understand.
Authors’ response: Thank you. Results and discussion section have been changed according to your advice.
Revewer: All tables and figures should be deeply revised (for punctuation, format, and clarity).
In terms of format and writing, punctuation should be checked in the entire manuscript. References in the text do not maintain the same format (sometimes, there is a white space before and after the brackets, and this is not consistent throughout the manuscript).
Authors’ response: Thank you. All tables and figures were replaced. Formats, punctuation, and references were checked again and modified in the entire manuscript.
SPECIFIC COMMENTS:
INTRODUCTION
Reviewer: ***It is a well-structured Introduction section. However, it contains too many parts with specific information about studies, which increases the extension of this section. It is recommended to include information in a general way, and to avoid giving details from studies. Lines 58 to 74 and 75 to 84 are confusing. These paragraphs have information about different aspects that can be improved with Tai Chi, without being linked. This fact makes that these paragraph lacks of fluency. It is recommended to re-organize this information according to the key aspects (those really related with postural control such as Time up and go, etc).
Authors’ response: Thank you for the comment. These paragraphs were modified as follows (Line 50 – 66):
Tai Chi has been proposed as one of the effective therapies for partially restoring neuromuscular function in people with PN. Li and coworkers [15] observed that choosing an appropriate exercise can improve the damaged sensory system, restore activity and postural control, and reduce the PN patients’ dependence on others. Tai Chi training restores postural control by reducing neuropathy total symptom score (TSS) [24], delaying proprioceptive declination by improving somatic sensory sensitivity[25], improving proprioception [26], and reducing the neuromuscular reaction time of the lower extremity muscles [27]. People who regularly practiced Tai Chi could maintain a faster reaction time of postural muscles during regressive walking [28] and down-step walking [29].
Tai Chi training can improve postural control in people with PN. It is part of the comprehensive rehabilitation of people with PN, closely related to functional rehabilitation. In recent years, more researchers have paid attention to the role of Tai Chi in improving static postural control [24, 25, 30 – 36] and dynamic postural control [12, 14, 25, 30, 34, 36, 37] of people with PN. But the results are still controversial. For example, the time of one-leg standing test (OLST) is a good indicator for static postural control. It showed that OLST may be longer after Tai Chi training [24], but the others got different results [30, 35 – 37].
Reviewer: Line 85: This sentence is unclear, please reword it.
Authors’ response: Thank you for the comment. This sentence was corrected to the following: Although there are many studies investigating the effects of Tai Chi on postural control in people with PN, there is a lack of relevant systematic reviews. (Line 67 – 69).
Reviewer: Line 87: “the efficacy of Tai Chi on people with DPN”, but, what exactly? postural control of these people?
Authors’ response: Thank you. This sentence was modified more clearly. “Previous systematic reviews concentrated on the postural control benefits of Tai Chi, focusing on the non-PN population.” (Line 69 – 70)
Reviewer: Also, it would be appropriate to focus on the importance of postural control for this pathology in this section, and those tests/functions that authors considered as postural control.
Authors’ response: Thank you for the comment. We modified the introduction section to clarify.
METHODS
Reviewer: PubMed is not a database.
Authors’ response: Thank you. We used PubMed to search the Medline database. Changes were made accordingly. (Lines 89 – 92)
Reviewer: Why February 2022? why not the entire 2022 year? It is recommended to consider this aspect to have an updated systematic review. If this systematic review is published, it would be useful to be as up to date as possible (at least, up to the end of the last year).
Authors’ response: Yes, we conducted a second search. It is up to date now.
Reviewer: Lines 112 – 113: Since this is a systematic review, it would be more appropriate include “studies that recruited participants according to the following criteria…” because this work did not really include participants.
The same for the exclusion criteria.
Authors’ response: Thank you very much and we characterized the participants in this part (Line 107 – 112). “Participants with a clinical diagnosis of PN based on symptoms have not practiced Tai Chi in the past four months before trials; They had an increase in vibrating perception threshold (VPT), a decreased nerve conduction velocity (NCV), and engaged in moderate intensity exercise at least once a week; The body mass index (BMI) was between 10 and 35kg/m2.”
Reviewer: Intervention: why 8 weeks? to include a reference would be useful to understand the minimal duration that authors proposed.
Authors’ response: Thank you very much. We consider this duration as a basic adaptation to Tai Chi. No studies set training duration as less than 8 weeks. So we removed the criteria (Lines 113 – 115).
Reviewer: Outcome measures: I do not understand why authors considered the PPDT test as a postural control test. This test is highly different from the rest of test considered (all of which inform about gait function or postural stability). Re-consider this inclusion criteria to maintain the coherence of the test considered as eligible.
Authors’ response: Thank you very much. Initially, we wanted to explain the mechanism of the Tai Chi effect on postural control. For the purpose to make the assessment focusing on postural control, we removed the PPDT section according to your advice in the revised manuscript.
Reviewer: Also, all tests included should be explained with more details in order to be able to understand the Results section.
Authors’ response: Thank you. In lines 119 – 124, we added some explanation of those indicators. And we add more explanation in the result section (Line 363 – 364, 370 – 372, 383 – 384, 393 – 395).
Reviewer: Study design: RCTs need to be explained the first time it appears in the text (the abbreviation has been explained in a posterior paragraph, please, check it).
Authors’ response: Thank you. We corrected it and checked all the abbreviations (Line 18, 117).
Reviewer: Lines 138 – 144: This paragraph does not add any information. Exclusion criteria cannot be the opposite to the inclusion criteria. If there is no exclusion criteria to highlight, it is recommended to delete this paragraph.
Authors’ response: Thank you. We reorganized this section according to your advice (Lines 126 – 128). “These studies were excluded if: the full-text article could not be retrieved in English language and Chinese language, and the articles were animal experiments, meeting abstracts, case reports, methodological experimental designs, or reviews.”
Reviewer: Data extraction procedure should be detailed. What researches? was this screening carried out independently? what if disagreement among both researches? who verified this information? Also, it should be detailed what information was extracted: causes of loss to follow-up, results, conclusions, etc is not correct. What data was extracted for the MA?
Authors’ response: Thank you. We reorganized this section according to your advice (Lines 129 – 146).
“Two trained reviewers used a multi-step screening approach to assess the included studies independently and recorded the main reason for exclusion. The author, publication year, and title were reviewed in the first screening to determine including studies. In the second screening, titles, and abstracts were reviewed to exclude studies that did not meet the inclusion criteria. In the final screening, the full texts were reviewed to determine the outcome indicators and other relevant data such as author, publication year, sample size, selection criteria, intervention measures, frequency and intensity, intervention duration, follow-up time, causes of loss to follow-up, results, and conclusions.
Disagreements were resolved through discussion. And the final decision was made by a third author if consensus can’t be reached by discussion. This data was then extracted for use in the meta-analysis by one author independently and was confirmed by another author.
In addition, the literature characteristics of each included report were documented: reasons for duplicative systematic reviews, number of actual applications versus previous reviews, experimental methods of inclusion, demographic data, and characteristics, as well as heterogeneity and subgroup analysis of the main studies, and re-view conclusions on whether Tai Chi is more effective for balance function of people with PN.”
Reviewer: Lines 154 – 169: On what reference are based the procedure to established the risk of bias classification? In this sense, many tools specifically designed to evaluate risk of bias of RCT are available in the literature (i.e. those tools proposed by Cochrane). Please, consider to use it. Pedro scale has 11 items (10 items for the score), why authors used only 8 items? Again, what happened in case of disagreement? it is not clear in the text. Please, re-arrange information included to explain the risk-of-bias evaluation. A reference is needed.
Authors’ response: Thank you very much. Outcomes of the risk of bias classification conducted by the RevMan 5.3.0. It is not suitable to evaluate the risk of bias of RCT by parts of the Pedro scale. We evaluated the risk of bias of RCT again by the Cochrane Collaboration’s tool for assessing the risk of bias (updated 2022) and added a reference (Line 148 – 162).
41 Higgins J., Savović J., Page M., Elbers R., Sterne J. Chapter 8: assessing risk of bias in a randomized trial. In: Cochrane Handbook for Systematic Reviews of Interventions Version 6.3.; 2022.
RESULTS
Reviewer: This section needs to be restructured and well-organized. Now, text, tables and figures are difficult to understand and follow because the lack of organization.
Authors’ response: Thank you very much. We re-structured sections. First, we updated the flowchart according to PRISMA2022; Second, we combined figures into three new figures-static postural control of comparison between groups, dynamic postural control of comparison between groups, and dynamic postural control compared between after and baseline. Third, we reorganized the text (Line 179 – 314).
Reviewer: Figure 1: Quality of the figure is poor. Some sentences in the boxes begin with lowercase, please, check it. In many brackets, there are white spaces that should be checked. Also, it is recommended to modify figure letter type to maintain the same format in the entire manuscript. There are abbreviation without explaining.
Authors’ response: Thank you. The quality of Figure 1 has been improved.
Reviewer: It is recommended not to start the RESULTS heading with a figure that has not already been referenced in the text. Instead, add the referencing text.
Authors’ response: Thank you. Revised according to your advice.
Reviewer: This text should not duplicate the figure 1. Please, modify it and include the essential data or trends to highlight.
Authors’ response: Thank you. This section has been edited ( Lines 179 –195).
Reviewer: Lines 197 – 198: This information has been included in the previous paragraph, delete it.
Authors’ response: Thank you. It has been deleted (Lines 200 – 202).
Reviewer: Basic characteristics of included studies: How many females and males? age? duration of interventions? type of exercises as control intervention? Despite the fact that this information is detailed in the table, authors should highlight those important aspects in the text. What about the locations of the investigations? It is useful to know where these experimental studies have been carried out (highlighting the most repeated locations, for example).
Authors’ response: Thank you. This section has been revised according to your advice(Lines 179 – 195).
Reviewer: Table 1 needs a deep revision of the punctuation and other important aspects (spaces between signs, abbreviations without explaining, etc). Take into account that Table should be understandable by itself.
Authors’ response: The format of Table 1 has been revised and explanatory notes have been added to Table 1.
Reviewer: Line 203: the first sentence can be deleted (since it belongs to Methods). Also, it is recommended to simplify this paragraph, now it is difficult to read (and it is duplicating information of Figures 2 and 3).
Authors’ response: Yes, it has been deleted (Line 204). The reverent text has been simplified (Lines 204 – 211).
Reviewer: Lines 233 – 236: This information should be moved to Methods section.
Authors’ response: Yes, it has been deleted (Line 223).
Reviewer: Lines 236 – 238: Again, this information belongs to statistical analysis.
Authors’ response: Thank you. We included this part to meet the requirement of PRISMA 19, 20b, and 20d.
PRISMA: 19, For all outcomes, present, for each study: (a) summary statistics for each group (where appropriate) and (b) an effect estimate and its precision (e.g. confidence/credible interval), ideally using structured tables or plots.
20b, Present results of all statistical syntheses conducted. If meta-analysis was done, present for each the summary estimate and its precision (e.g. confidence/credible interval) and measures of statistical heterogeneity.
20d, If comparing groups, describe the direction of the effect. resent results of all sensitivity analyses conducted to assess the robustness of the synthesized results.
Reviewer: Lines 233 – 247: The entire paragraph should be rearrange to make easier to follow. Authors should include only that information about results, by avoiding to include information about what statistical analysis were completed or what are the meaning of the test or results.
Authors’ response: Thank you. It has been reorganized (Lines 222 – 239).
Reviewer: It is recommended to join figures 4 to 11 in an only figure with different parts (because all these figures represent Forest plots of the MA). It would be easier to understand and paper structure would improve.
Authors’ response: Thank you. Reconfigured as suggested.
DISCUSSION
Reviewer: This should be organized by following the same information previously included in the manuscript. If Methods section did not include postural control test as “dynamic” or “static”, why authors included these subheading in the discussion?
Authors’ response: Thank you for the comment. We reconstructed the Methods, Result, and Discussion sections, and tried to keep them consistent. In the Methods section, we added an explanation to outcome measures; In the Result section, the titles have been changed to add “static postural control” and “dynamic postural control”. (Line 222, 253).
Reviewer: ***Please, try to maintain the consistency in the entire manuscript. Therefore, the entire discussion should be reorganized to be consistent with the previous part of the manuscript. Now, it is difficult to understand.
Authors’ response: Thank you. Methods, Result, and Discussion sections have been reconstructed.
Reviewer: Format should be revised. Punctuation should be checked in the entire manuscript. References in the text do not maintain the same format (sometimes, there is a white space before and after the brackets, but it is not consistent throughout the manuscript). Please, check it.
Authors’ response: Thank you. All tables and figures were replaced. Formats, punctuation, and references were checked again and modified in the entire manuscript.
Reviewer 3 Report
I had the opportunity to review this interesting article on the effects of Tai Chi on peripheral neuropathy. Although the idea is relevant and novel for a systematic review, I have some concerns regarding the methodological aspects of the study. First, some parts of the methods are not described sufficiently (e.g., search strategy). Second, reporting bias and certainty of evidence are not evaluated in the study, which is a major limitation. Third, my main concern is that the study aimed to evaluate the effects of Tai Chi on postural balance in diabetic patients with peripheral neuropathy. However, studies evaluating the effects of Tai Chi on peripheral neuropathy in general, and not specifically in patients with diabetes, are also included in the review and the meta-analysis.
I have explained more about my concerns below.
Major comments
a. I would suggest rewriting and reorganizing the methods section according to the PRISMA 2020 guideline.
b. Have you registered your study with Prospero or any other registry?
c. Please describe your full search strategy. I would suggest adding the search strategy for all databases as a supplementary file.
d. When did you search the databases? Please mention the exact date.
e. Line 122; Please provide some examples of what you meant by severe disease.
f. The main objective of your study was to evaluate the effects of Tai Chi on individuals with DPN. However, your eligibility criteria for participants don’t contain any criteria on how you defined a patient with diabetes. Also, some of the included studies evaluated the effects of Tai Chi on PN in general and not specifically in individuals with diabetes (e.g., Brad et al., Hermanns et al., and Tousignant et al.).
g. Line 153; Please mention all extracted variables and avoid using “etc.”
h. Did two researchers screen the articles independently? If so, how did they resolve the conflicts?
i. Please use the 2020 Prisma flow diagram.
Minor comments
a. Lines 58-74; I would suggest summarizing this paragraph and only focusing on the beneficial effects of Tai Chi in individuals with DPN.
b. Line 76; Only Xu et al.’s study has a cross-sectional design. Please revise the sentence and correct it
Author Response
Major comments
Reviewer: a. I would suggest rewriting and reorganizing the methods section according to the PRISMA 2020 guideline.
Authors’ response: Thank you very much. The methods section has been rewritten and reorganized according to your advice and PRISMA 2020.
Reviewer: b. Have you registered your study with Prospero or any other registry?
Authors’ response: Thank you. This systematic review was registered with INPLASY (International Platform of Registered Systematic Review and Meta-analysis Protocols, INPLASY202340098) instead of Prospero. This information is now added to the first sentence of section 2.1.
Reviewer: c. Please describe your full search strategy. I would suggest adding the search strategy for all databases as a supplementary file.
Authors’ response: Thank you very much. We searched the database again to make sure that new studies are meeting our included criteria. Then we provided supplementary material of the Literature search strategy this time. And in lines 87 – 96 of the revised manuscript we updated the searching time, added the details of the search equation, and filter.
Reviewer: d. When did you search the databases? Please mention the exact date.
Authors’ response: The first search was on February 2022. The latest search was on 28th April 2023(Line 89).
Reviewer: e. Line 122; Please provide some examples of what you meant by severe disease.
Authors’ response: Thank you. According to comments we rewrote this section with a more concise approach. And we will pay attention to this problem in the future.
Reviewer: f. The main objective of your study was to evaluate the effects of Tai Chi on individuals with DPN. However, your eligibility criteria for participants don’t contain any criteria on how you defined a patient with diabetes. Also, some of the included studies evaluated the effects of Tai Chi on PN in general and not specifically in individuals with diabetes (e.g., Brad et al., Hermanns et al., and Tousignant et al.).
Authors’ response: Thank you very much. We are very sorry for this misunderstanding between the authors. We checked the former manuscripts and data. And we corrected this mistake to make sure that the population is people with PN instead of DPN in the revised manuscript. We are very sorry again.
Reviewer: g. Line 153; Please mention all extracted variables and avoid using “etc.”
Authors’ response: Thank you. The “etc” was deleted. And we added “selection criteria”. (Lines 136 – 137)
Reviewer: h. Did two researchers screen the articles independently? If so, how did they resolve the conflicts?
Authors’ response: Yes. We added this description on Lines 130 – 141.
Reviewer: i. Please use the 2020 Prisma flow diagram.
Authors’ response: Thank you very much. The flow diagram has been replaced and reorganized according to your advice and PRISMA 2020.
Minor comments
Reviewer: a. Lines 58-74; I would suggest summarizing this paragraph and only focusing on the beneficial effects of Tai Chi in individuals with DPN.
Authors’ response: Thank you for the comment. These three paragraphs were summarized into two paragraphs (Lines 50 – 66).
Reviewer: b. Line 76; Only Xu et al.’s study has a cross-sectional design. Please revise the sentence and correct it
Authors’ response: Thank you. “several” has been deleted and the sentence is rewritten as “Tai Chi training restores postural control by reducing neuropathy total symptom score (TSS) [24]” (Lines 53 – 54).
Round 2
Reviewer 2 Report
Dear Authors,
Thank you for a very good job in answering my questions and recommendations. The authors did a very good job, and the paper quality increased accordingly to the effort put into revision.
Best regards,
Author Response
Thank you.
Reviewer 3 Report
Thank you very much for giving me the opportunity to review this interesting manuscript again. I would like to thank the authors for making extensive revisions to the article. However, I still have serious concerns about the methodology of the study and the validity of the findings. There are some disparities between the methods and results sections. For example, on page 3, line 132, it is mentioned that Medline was searched in this study. Yet there is no information on the number of articles retrieved by searching Medline in the flow diagram. Also, it is mentioned in the methods section that only RCTs were included in the review. However, ineligible studies are included in the review. Brad Manor et al.’s study had a non-controlled and non-randomized design. I also wonder how authors have judged this study as having a low risk of bias in the random sequence generation domain. Elisabeth I. Cavegn et al.’s study had a single arm. In fact, the referent group consisted of healthy adults who were used as a basis for comparison with individuals with diabetes. Janusz Maciaszek et al.’s study was performed on older adults with dizziness, not individuals with PN. Again, Michel Tousignant. et al.’s study was performed on frail older adults, not individuals with PN.
Considering these major issues, I have serious concerns regarding the validity of the findings, and I still recommend against the publication of the manuscript.
Author Response
Thank you for your comments and suggestions. We have edited the manuscript based on your comments.
More specifically:
- “Medline” has been deleted.
- The theme of Brad Manor et al.’s study is very close to this meta-analysis and important to form our question. However, it is not RCT. We thought it over and decided to delete this study from the meta-analysis according to your comments (Table 1, Figure 5, and Figure 6).
- We agreed with you that healthy adults were used in the Cavegn et al., study as a basis for comparison with diabetes. On the other hand, they were also used as a basis for the Tai Chi effect on peripheral somatosensation in an imprecise way of comparison. The results of this study lowered the effect of Tai Chi on indicators of dynamic control (Figure 5, Figure 6). We kept this study in as a situation of current unoptimistic research on the Tai Chi effect on peripheral somatosensation.
- Thank you and we have excluded Maciaszek et al.’ and Tousignant. et al.’s studies. We are sorry for these mistakes and thanks so much for your careful screening.